# Perceptions of School Climate and Internet Gaming Addiction among Chinese Adolescents: The Mediating Effect of Deviant Peer Affiliation

**DOI:** 10.3390/ijerph19063604

**Published:** 2022-03-18

**Authors:** Hongyu Zou, Yuting Deng, Huahua Wang, Chengfu Yu, Wei Zhang

**Affiliations:** 1School of Psychology, South China Normal University, Guangzhou 510631, China; zouhongyu2016@163.com (H.Z.); zhangwei@scnu.edu.cn (W.Z.); 2College of Liberal Arts and Social Sciences, City University of Hong Kong, Hong Kong 999077, China; 3School of Education, Guangzhou University, Guangzhou 510006, China; 15767364387@163.com (Y.D.); 2112108032@e.gzhu.edu.cn (H.W.)

**Keywords:** adolescent, perceptions of school climate, internet gaming addiction (IGA), deviant peer affiliation

## Abstract

Adolescent internet gaming addiction (IGA) has become a serve public health problem, particularly in China. Despite the fact that the school climate has a significant impact on teenage IGA, little research has looked into the underlying mediating processes. This cross-sectional study looked at the impact of adolescents’ perceptions of their school climate (including teacher–student support, student–student support, and opportunities for autonomy) on IGA in a sample of 1053 Chinese adolescents (*Mean*_age_ = 13.52 years; 52.4% females) using convenient sampling methodology, as well as whether deviant peer affiliation mediated this association. Adolescents’ school climate, IGA, and deviant peer affiliation were examined using the School Climate Questionnaire, Internet Gaming Addiction Scale, and two validated tools in this study. The results showed that teacher–student support and student–student support were both negatively associated with IGA, and that these correlations were mediated by deviant peer affiliation. The implications of these findings are discussed.

## 1. Introduction

According to the China Internet Network Information Center’s 47th Statistical Report on China’s Internet Development, students made up the largest group of Internet users, accounting for 21% as of December 2020. Internet gaming addiction (IGA) is a specific subtype of Internet addiction characterized by an addiction to playing online video games, role-playing games, or any other interactive gaming environment accessible over the Internet [1]. An increasing body of research suggests that teenage IGA is linked to a number of negative developmental outcomes, including poor academic performance [2], depression [3], social anxiety [4], violent behavior [5], or even suicide [6]. Therefore, it is critical to pay close attention to the process of teenage IGA generation and to establish protective intervention strategies.

### 1.1. Perceptions of School Climate and Adolescent IGA

Adolescent development theories such as stage–environment fit theory [7], the developmental assets model [8], and the positive youth development theory [9] all agree that a pleasant school environment is critical for adolescent development. School climate, as a basic component of the learning environment, has received more attention in the previous decade [10,11].

Jia et al. [10] found that adolescents’ perceptions of school climate are primarily influenced by their perceptions of emotional and academic support from teachers (teacher–student support), emotional support among students (student–student support), and opportunities for choice and decision making in terms of learning and classroom life (opportunities for autonomy in the classroom). There is a lot of evidence showing students’ assessments of three characteristics of school climate are strong predictors of their physical, psychological, and academic results [12]. In addition, there are several studies investigated the impact of school climate on adolescent IGA [13,14,15]. For example, Chang and Kim [14] reported that poor school climate is a substantial risk factor for adolescent gaming addiction. Similarly, Juthamanee and Gunawan [16] discovered that teenage perceptions of school climate (friendship support) were associated with Internet and video game addiction. These data imply that a positive school climate is critical for teenage IGA prevention and reduction.

### 1.2. Deviant Peer Affiliation as a Mediator

However, past research has focused on the direct impact of school climate impressions on adolescent online gaming addiction. On IGA, no one knows how the school climate works. This study intends to investigate this underlying mediation mechanism in order to better understand the genesis, prevention, and treatment of teenage IGA.

According to social learning theory [17], the impact of school climate on adolescent problematic behavior (including IGA) is mostly indirect, as it is mediated by primary socialization sources such as peers. In other words, deviant peer affiliation might act as a mediator in the IGA–school climate link. On the one hand, past research has established a link between school climate and deviant peer affiliation. Wang et al. [18], for example, discovered that a favorable school climate was connected with deviant peer affiliation. Adolescents who experience a poor school climate have a weak association to their schools, according to Li et al. [19], which makes them less likely to feel guilty when they engage in hazardous activities with deviant peers. Deviant peer affiliation, on the other hand, is one of the primary influencers for teenagers to engage in gaming [6], since it produces peer pressures that cause adolescents to engage in deviant activities [20]. Deviant peer affiliation was also found to play a mediating function in the connection between school setting and adolescent Internet activity in empirical investigations. For example, Li et al. [19] discovered that deviant peer affiliation mediated the relationship between perceived school climate and adolescent Internet addiction. Jia et al. [20] also discovered that deviant peer affiliation mediated the connection between the teacher–student relationship and Internet addiction. As a result of these findings, it is plausible to assume that deviant peer affiliation will likewise mediate the influence of teenage views of school atmosphere on IGA.

### 1.3. The Present Study

We wanted to see if deviant peer affiliation mediated the influence of teenagers’ evaluations of school climate (i.e., teacher–student support, student–student support, possibilities for autonomy in the classroom) on their IGA in order to fill in some gaps in the research. We offer the following two hypotheses, both of which are represented in Figure 1, based on the material reviewed above:

**Hypothesis** **1:** *Teacher–student support, student–student support, and classroom autonomy possibilities will all be negatively correlated with adolescent IGA*.

**Hypothesis** **2:** *Teacher–student support, student–student assistance, possibilities for autonomy in class, and teenage IGA will all be mediated by deviant peer affiliation*.

## 2. Method

### 2.1. Participants

A total of 1053 teenagers (52.4% females) were recruited from 4 cities in China’s Guangdong Province, using convenient sampling methodology. They were in Grades 6–7 and were between the ages of 11 and 16 (*M**ean*_age_ = 13.52 years, *SD* = 0.79). Approximately 88.8% of the participants were from intact families, 6.6% from non-intact families, and 3.3% from other families (1.3% did not disclose). In total, 54.3% of their fathers and 63.1% of their mothers had less than a high school education, and 59.1% of the participants came from families with a per capital monthly income of less than CNY 3001.

### 2.2. Procedures

The assessment was conducted in class. Data were collected by well-trained graduate students in psychology from South China Normal University and mental counselors from the schools being surveyed (all of whom had a college degree). The participants took about 45 min to complete the whole evaluation. After the surveys were completed, they were immediately returned. The Ethics in Human Research Committee of South China Normal University authorized all materials and methods.

### 2.3. Measures

#### 2.3.1. Perceptions of School Climate

The 25-item School Climate Questionnaire [10], which assesses three dimensions, teacher–student support (e.g., “teachers believe I can do well”), student–student support (e.g., “students care about one another”), and opportunities for autonomy in the classroom (e.g., “students are given the chance to help make decisions”), was used to assess adolescents’ perceptions of school climate. Each subscale of this questionnaire exhibited high reliability and validity for both American and Chinese teenagers, according to Jia and colleagues [10]. On a 4-point scale ranging from 1 = never to 4 = often, adolescents stated how often they experienced each of the 25 items. Each subscale’s total score was calculated, with higher scores indicating more support or possibilities for autonomy in the classroom. The Cronbach’s α coefficients for the three subscales in this study were 0.85, 0.87, and 0.73, respectively.

#### 2.3.2. Deviant Peer Affiliation

The average level of 10 questions modified from 2 validated measures [21] was used to assess adolescent affiliation with deviant peers. On a 5-point scale ranging from 1 = none to 5 = 5 or more, adolescents were asked how many of their friends smoked cigarettes, drank alcohol, skipped or cut school, cheated on school tests, stole, were physically and verbally aggressive, had Internet addiction, received school punishment, had gone to a bar or night club, and watched porn movies or books in the previous 6 months (e.g., “How many of your friends smoked cigarettes in the last six months?”). The higher the total score of the 10 items, the more deviant the peer association. The Cronbach’s α coefficient in this study was 0.82.

#### 2.3.3. IGA

The 11-item Internet Gaming Addiction Scale [22] was used to measure adolescents’ IGA. On a 3-point scale, adolescents indicated how often this occurred in the previous 6 months for each of the 11 items (e.g., “Have you tried to play online games less often or for shorter periods of time, but have been unsuccessful?”). Never = 0, sometimes = 0.5, and yes = 1 were recoded, and the total score of 11 items was calculated, with higher scores indicating greater IGA. The Cronbach’s α coefficient in this study was 0.69.

### 2.4. Statistical Analysis

SPSS for windows version 20.0 (IBM Corporation, Armonk, NY, USA) was used for all statistical analyses. The frequency of IGA and the demographic parameters of the students were described using descriptive analysis. The mediating impact of deviant peer affiliation in the connection between school climate and adolescent IGA was investigated using multiple regression analysis. The statistical significance criteria was set at *p* < 0.05.

## 3. Results

### 3.1. Prevalence of IGA

According to the Internet addiction diagnostic criteria proposed by Young in 1996 [23], an adolescent who exhibited at least five criteria on the IGA questionnaire was considered as having IGA. In the current sample, 7.69% (*N* = 81) of the participants in the current sample displayed IGA.

### 3.2. Descriptive Analyses

The means, standard deviations, and bivariate correlations between the variables in the current study are presented in Table 1. Table 1 shows that three subscales of school climate perceptions had moderate significantly positive associations with each other, which is consistent with earlier research in Chinese teenagers [10]. Three subscales of school climate perceptions were significantly inversely related to deviant peer affiliation and IGA. IGA was shown to be strongly linked to deviant peer affiliation.

### 3.3. Testing for Mediation Effect

To test Hypothesis 2: Teacher–student support, student–student assistance, possibilities for autonomy in class, and teenage IGA will all be mediated by deviant peer affiliation, we utilized MacKinnon’s four-step process [24], which is based on the most extensively used method for evaluating mediation [25,26]. First, there must be a significant relationship between school climate and to IGA. Second, there is a significant relationship between deviant peer affiliation and perceptions of school climate. Third, deviant peer affiliation must be significantly related to IGA when both perceptions of school climate and deviant peer affiliation are predictors of IGA. Fourth, the coefficient for the indirect path between perceptions of school climate and IGA via deviant peer affiliation must be significantly. The Sobel test is used to see if the last step is satisfied. Gender, age, household per capital income (from 1 = 0–1000 CNY to 10 = 9001 CNY or more), and father’s and mother’s educational level (from 1 = Primary or below to 7 = Doctor degree or above) were all included as covariates in all analyses.

As shown in Table 2, regression analyses indicated that teacher–student support (*b* = −0.12, *p* < 0.05) and student–student support (*b* = −0.37, *p* < 0.001) were both significantly associated to IGA in the first stage, but possibilities for autonomy were not significantly related to IGA (that is, Hypothesis 1 was partially supported). In the second step, teacher–student support (*b* = −0.12, *p* < 0.05) and student–student support (*b* = −0.37, *p* < 0.001) were significantly related to deviant peer affiliation. In the third step, when both perceptions of school climate and deviant peer affiliation were predictors of IGA, deviant peer affiliation was significantly related to IGA, *b* = 0.77, *p* < 0.001. Finally, the Sobel test showed that the indirect effect of teacher–student support and student–student support on adolescent IGA via deviant peer affiliation were significant, *Z*_1_ = −2.02, *Z*_2_ = −4.41, *ps* < 0.05. In conclusion, all of the four conditions for establishing mediation effect were fully satisfied, with the exception of autonomy opportunities. Therefore, Hypothesis 2 was partially supported.

## 4. Discussion

IGA is becoming more of a concern among adolescents, particularly in China. Because of the gravity of this issue, we must investigate the elements that influence it and devise effective preventative and intervention strategies. The effect of perceptions of school climate (including teacher–student support, student–student support, and opportunities for autonomy in the classroom), one of the most influential socialization domains in an adolescent’s life, on adolescent IGA was investigated, as well as the underlying mechanism. Findings indicating that deviant peer affiliation explained a portion of the protective impact of school climate views on teenage IGA. We described the prevalence of IGA in Chinese teenagers before testing the aforementioned model. The current sample had 7.69% of participants who were addicted, which is consistent with the previous researches of “rates of possible IGD among Adolescents in China ranged between 2% and 17%” [27,28,29,30]. It is possible that this is due to the fact that the individuals in the current study are younger than those in the study by Yu and colleagues [27]. According to a review, the prevalence of IGA is greater in older individuals than in younger people [31].

First, while controlling for gender, age, family income, and father and mother’s educational level, we discovered that two dimensions of school climate perceptions, teacher–student support and student–student support, were significantly negatively associated with adolescent IGA, which is consistent with Hypothesis 1. These findings reflect developmental theories that imply emotional and academic support from both instructors and students are essential for teenage adjustment [7,9]. Unexpectedly, there is no correlation between possibilities for autonomy in the classroom and IGA. In addition, Jia and colleagues (2009) discovered that possibilities for autonomy in the classroom had no specific influence on teenagers’ self-esteem or sadness [10]. Alternatively, it is possible that options for autonomy in the classroom are just discussed rather than implemented because it cuts down on class time. Nonetheless, a large body of empirical research shows that meeting autonomy needs in real life is a key protective factor in preventing Internet addiction [32] and that attempting to meet autonomy needs is a substantial risk factor for Internet addiction [31]. As a result, it is possible that the classroom is not the primary source of autonomy, resulting in lower IGA. Future research should look at the impact of possibilities for autonomy in other areas (such as family) on adolescent IGA.

Second, we found that deviant peer affiliation is a key underlying mechanism that helps interpret the association between perceptions of school climate (teacher–student support and student–student support) and adolescent IGA. In other words, when adolescents perceive positive teacher–student and student–student support, they are less likely to associate with deviant peers, which is related to less IGA. These findings support the social learning hypothesis [17], as well as previous research that found that deviant peer affiliation served as a crucial “bridge” between the school milieu and adolescent Internet addiction [19,20]. Adolescents in a favorable school atmosphere, on the other hand, may receive greater teacher and student support, which will satisfy their security, support, and intimacy requirements. As a result, they will be less likely to associate with deviant peers and participate in problematic Internet use, allowing IGA to grow.

### Limitations and Future Directions

Several limitations also important to noted when interpreting the results of this study. To begin with, the cross-sectional design precludes any causal conclusions. Longitudinal studies are needed to further understand the causal relationship. Second, all of the data were gathered through teenagers’ self-reports. Despite the fact that self-reported IGA ratings are typically regarded as credible [33], it is possible that teenage self-reports on deviant peer affiliation are exaggerated. As a result, future research should collect data on deviant peer affiliation using several sources (e.g., teacher report, peer self-report). Third, the psychological well-being of teenagers was not examined. Future research should look at the impact of teenage psychosocial issues including anxiety and sadness on IGA. Meanwhile, the duration of adolescents’ Internet addiction has not been investigated. The mechanism of Internet addiction for lifetime gamers and past-year gamers may differ to some extent, and future research should focus on this question. Finally, the current research is confined to teenagers in China’s Guangdong Province. Because of the population’s subgroup uniqueness, future research should test the current finding in different cultural and/or geographical circumstances.

Nonetheless, the findings of the study show that teacher–student and student–student support were important protective factors for adolescent IGA, and that these associations, which were mediated by deviant peer affiliation, were extremely useful for guiding prospective and developing intervention practices and programs. The following are some of the study’s specific implications for future research, preventative measures, and program development: (1) Further research is needed to examine the impact of opportunities for autonomy in multi-level contexts other than the classroom on adolescent IGA; (2) prevention programs of promoting school climate, especially for teacher–student support and student–student support, could be effective in reducing the risk of adolescent IGA; (3) prevention programs reducing deviant peer affiliation, particularly among adolescent who report lower levels of perceptions of school climate, could also be effective in reducing the risk of adolescents involve in IGA.

## 5. Conclusions

The present study extends the adolescent development theories [7,8,9] and the social learning theory [17] to the study of IGA by testing the mediating role of deviant peer affiliation. Previous studies have studied perceived school climate as a general part of this [19]. Our study further explores how different dimensions of perceived school climate play a protective role in IGA. The current results enrich our understanding of the mechanisms underlying the relationship between school climate and IGA among Chinese adolescents. Our findings revealed that teacher–student support and student–student support but not opportunities for autonomy have the protective effect on the IGA through decreasing the chance of deviant peer affiliation. Therefore, teachers should give students more emotional support, practical problems guidance in academy and life. In addition, more activities to enhance class cohesion and enhance students’ awareness and opportunities to help each other will be held by school, thus effectively preventing adolescent IGD.

## Figures and Tables

**Figure 1 ijerph-19-03604-f001:**
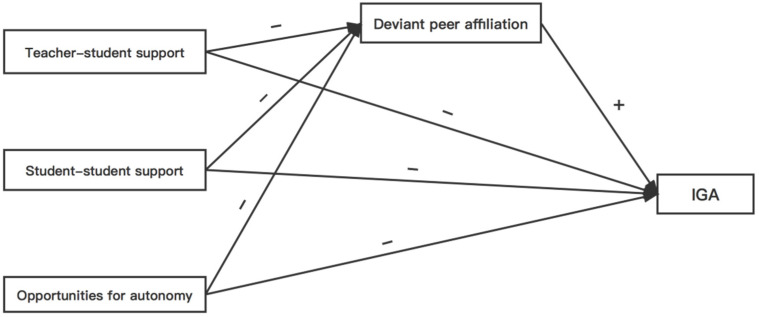
The conceptual model of perceptions of school climate, deviant peer affiliation and adolescent IGA.

**Table 1 ijerph-19-03604-t001:** Means, standard deviation, and bivariate correlations between perceptions of school climate, deviant peer affiliation, and adolescent IGA (*N* = 1053).

Variables	Mean	SD	1	2	3	4
1. Teacher–student support	3.67	0.77	-			
2. Student–student support	4.00	0.67	0.57 ***	-		
3. Opportunities for autonomy	3.25	0.88	0.67 ***	0.49 ***	-	
4. Deviant peer affiliation	1.09	0.22	−0.19 ***	−0.28 ***	−0.12 ***	-
5. Internet gaming addiction	0.17	0.16	−0.23 ***	−0.32 ***	−0.18 ***	0.27 ***

Note. *** *p* < 0.001.

**Table 2 ijerph-19-03604-t002:** Testing the mediation effect of perceptions of school climate on online gaming addiction (*N* = 1053).

	Step 1 IGA		Step 2 DPA		Step 3 IGA	
	*b*	*t*	*b*	*t*	*b*	*t*
CO: Gender	0.57	10.16 ***	0.18	2.95 **	0.54	9.74 ***
CO: Age	0.00	0.06	0.07	1.83	−0.01	−0.26
CO: Family income	0.00	−0.25	0.01	0.70	−0.01	−0.38
CO: Father’s educational level	−0.03	−0.71	0.08	1.94	−0.04	−1.07
CO: Mother’s educational level	−0.05	−1.35	−0.09	−2.13 *	−0.04	−0.99
X: Teacher–student support	−0.12	−2.21 *	−0.12	−2.15 *	−0.10	−1.86
X: Student–student support	−0.37	−7.15 ***	−0.37	−6.84 ***	−0.31	−5.90 ***
X: Opportunities for autonomy	0.02	0.52	0.08	1.78	0.01	0.21
ME: Deviant peer affiliation					0.77	5.80 ***
*R^2^*	0.19		0.10		0.21	
*F*	30.34 ***		14.15 ***		31.54 ***	

Note. Each column is a multiple regression equation that predicts the criterion at the top of the column. DPA means deviant peer affiliation. Gender was dummy coded such that 0 = female and 1 = male. CO means covariate; X means independent variable; ME means mediator; * *p* < 0.05. ** *p* < 0.01. *** *p* < 0.001.

## Data Availability

The data presented in this study are available on request from the corresponding authors (C.Y.).

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
