# Peer review of "Perceptions of School Climate and Internet Gaming Addiction among Chinese Adolescents: The Mediating Effect of Deviant Peer Affiliation"

_ijerph, 2022, doi:10.3390/ijerph19063604_

Round 1
Reviewer 1 Report
The authors explored deviant peer affiliation as a mediator between perceived school climate and IGA. My comments are as follows:
- Introduction, Deviant Peer Affiliation as a Mediator. The authors speculated about the role of deviant peer affiliation as a mediator instead of providing evidence from existing literature. For example, Social Learning Theory by Bandura, "antisocial patterns of behavior (including IGA)", antisocial is a strong word, it denotes psychiatric disorders that pose significant risks to others, IGA is not antisocial, it may be problematic at best. "In other words, great school climate may reduce(s) deviant peer affiliation, which, in turn...", the authors speculated about the connection between school climate and IGA here. The authors may need to review evidence from existing studies.
- Results. "According to Young's diagnostic criterion for Internet addiction."??? This is a mistake that should be corrected.
- Sample characteristics need to be presented.
- Discussion. "However, few researches explore its(') influencing factors and the underlying mediating mechanism." I do not agree with the authors on this, there are lots of studies from across the world on the etiology, and correlates of IGA, and many of these studies performed mediation analyses.
- Discussion. "This question is particularly urgent because demonstrating the determinants of IGA are the first step for develop(ing)..." I believe that the authors used a cross-sectional design, thus, the authors did not exactly examine the determinants of IGA, but rather its correlates.
- Discussion. The last paragraph in Discussion was a bit off, the authors speculated about the chained mechanism linking school climate and IGA via deviant peer affiliation, instead of actually assessing it in their research.
- Moderate English edits are needed.
Author Response
_Introduction
Comment 1. Introduction, Deviant Peer Affiliation as a Mediator. The authors speculated about the role of deviant peer affiliation as a mediator instead of providing evidence from existing literature. For example, Social Learning Theory by Bandura, "antisocial patterns of behavior (including IGA)", antisocial is a strong word, it denotes psychiatric disorders that pose significant risks to others, IGA is not antisocial, it may be problematic at best. "In other words, great school climate may reduce(s) deviant peer affiliation, which, in turn...", the authors speculated about the connection between school climate and IGA here. The authors may need to review evidence from existing studies.
Response: We are extremely grateful to reviewer for pointing out this problem. First, we have added some empirical studies to provide evidence in the elaboration of the mediating role of deviant peer affiliation (See in the Deviant Peer Affiliation as a Mediator section for details). Second, according to the reviewer’s comment, we have change “adolescent antisocial patterns of behavior (including IGA)” into “adolescent problematic behavior (including IGA)”.
_Results
Comment 2. Results. "According to Young's diagnostic criterion for Internet addiction."??? This is a mistake that should be corrected.
Response: We are extremely grateful to reviewer for pointing out this problem. We have made modifications as follows.
According to the Internet addiction diagnostic criteria was presented by Young in 1996 [23], an adolescent who exhibited at least 5 criteria on the IGA questionnaire was considered as having IGA. In the current sample, 7.69 % (n = 81) of the participants in the current sample displayed IGA.
Comment 3.Sample characteristics need to be presented.
Response: Thank you for the suggestion. We have added the Sample characteristics in the Participants section.
_Discussion
Comment 4.Discussion. "However, few researches explore its(') influencing factors and the underlying mediating mechanism." I do not agree with the authors on this, there are lots of studies from across the world on the etiology, and correlates of IGA, and many of these studies performed mediation analyses.
Response: We are extremely grateful to reviewer for pointing out this problem. We have deleted the sentence that reviewer mentioned and made modifications as follows: IGA is becoming a growing problem among adolescents, especially in China. The seriousness of this problem prompts us to explore its influencing factors and specify effective prevention and intervention programs.
Comment 5.Discussion. "This question is particularly urgent because demonstrating the determinants of IGA are the first step for develop(ing)..." I believe that the authors used a cross-sectional design, thus, the authors did not exactly examine the determinants of IGA, but rather its correlates.
Response: We are extremely grateful to reviewer for pointing out this problem. We have made modifications as follows: The seriousness of this problem prompts us to explore its influencing factors and specify effective prevention and intervention programs.
Comment 6.Discussion. The last paragraph in Discussion was a bit off, the authors speculated about the chained mechanism linking school climate and IGA via deviant peer affiliation, instead of actually assessing it in their research.
Response: We are extremely grateful to reviewer for pointing out this problem. According to the reviewer’s comment, we have modified expression in the last paragraph in Discussion based on our research results.
_Overall comments
Comment 7.Moderate English edits are needed.
Response: Thank you for your comment. We have reviewed and revised the language of this paper, which you can see in the latest version of the manuscript.
Reviewer 2 Report
This an interesting manuscript providing data regarding the relationship between school climate, deviant peers affiliation and Internet Gaming Addiction.
The study is well designed and data analysis is well structered.
I have only a few points, which could be considered as limitations or at least require a brief explanation.
1) The authors state clearly that the study was conducted in a specific province in China. This is however a limitations not only for the presence of cultural subgroups, but also because it is to proved that what is true for Chinese adolescents is true also for adolescents from other countries.
2) Psychological status of adolescents was not investigated. This is a relevant limitations, given the existing literature.
3) What was the time relation of the study with SARS-CoV-2 pandemia and with measures connected with the pandemia?
Author Response
Comment 1. The authors state clearly that the study was conducted in a specific province in China. This is however a limitations not only for the presence of cultural subgroups, but also because it is to proved that what is true for Chinese adolescents is true also for adolescents from other countries.
Response: Thank you for your comment. As suggested by reviewer, we have added the suggested content in the Limitations and Future Directions section. Specifically, “Finally, the present findings are limited to adolescents from the Guangdong Province, China. In view of subgroup specificity of population, future studies should examine the current conclusion in other cultural and/or geographical settings.”
Comment 2. Psychological status of adolescents was not investigated. This is a relevant limitations, given the existing literature.
Response: Thank you for your comment. According to the reviewer’s comment, we have added the limitation in the Limitations and Future Directions section. Specifically, “Third, psychological status of adolescents was not investigated. Future studies should consider the effect of psychological status of adolescents, such as anxiety and depression, on IGA.”
Comment 3. What was the time relation of the study with SARS-CoV-2 pandemia and with measures connected with the pandemia?
Response: Data of the present study were collected in 2019. Thus, there is no time relation with SARS-CoV-2 pandemia.
Reviewer 3 Report
Thank you for the opportunity of reading your manuscript. This is a relevant
relevant question and it is important that the authors address it. They have produced a well well-written manuscript. I think it is an interesting article and worth publishing, but I have a few comments that I would like the authors to address:
- Do you have any Funding?
- What are the bullet points of the study?
- The abstract is not present clear information about the methods. The authors must improve it.
- There is no sample size calculation.
- Did the authors asked about participants gender or sex? Please, make sure you reported in a right way.
- Please, improve the discussion, your results showed a lot of information that is not in the discussion topic.
- What are the strengths of the study?
- The authors are encouraged to check the formatting of the references cited as there are inconsistencies throughout the list and with the requirements for the journal. It is important to update the references (last five years), some of them are much old.
Author Response
Comment 1. Do you have any Funding?
Response: This study was supported by the National Natural Science Foundation of China [grant numbers 31600901].
Comment 2. What are the bullet points of the study?
Response: IGA is becoming a growing problem among adolescents. During adolescence, the influence of school on students’ IGD development becomes increasingly important. The current research mainly investigated three aspects of school environment: teacher-student support, student-student support and opportunities for autonomy influence on IGD via deviant peer affiliation. And we have found that peer affiliation is an important underlying mechanism which helps interpret the association between perceptions of school climate (teacher-student support and student-student support) and adolescent IGA.
Comment 3. The abstract is not present clear information about the methods. The authors must improve it.
Response: Thank you for the reviewer’s advice. School Climate Questionnaire, Internet Gaming Addiction Scale and 2 validated instruments were used to measured school climate, IGA and deviant peer affiliation of adolescents. The entire assessment took approximately 45 minutes for the participants to complete. We have add this details to abstract.
Comment 4. There is no sample size calculation.
Response: Thank you for your comment, according to the ratio of influencing factors and sample size of epidemiological investigation at least 50:1, 5 variables were included in this study, and about 250 sample size was expected to be required. Moreover, this study was a large sample study, so there was no problem of insufficient sample size.
Comment 5. Did the authors asked about participants gender or sex? Please, make sure you reported in a right way.
Response: Thank you for your suggestion. We make sure that there 52.4% females in the current study. The detail were introduced in the page 3.
Comment 6. Please, improve the discussion, your results showed a lot of information that is not in the discussion topic.
Response: Thank you for your critical and careful review. In the results, we showed the prevalence of the IGA, descriptive analysis of the interested variables and tested the mediating effect of deviant peer affiliation. We have further enrich our discussion, and details were in the page 5 and page 6.
Comment 7. What are the strengths of the study?
Response: Thank you for your suggestion. Previous studies have studied perceived school climate as a general part (Li et al., 2016; Ma, et al., 2017). Our study further explores how different dimensions of perceived school climate play a protective role in IGD.
The results showed that teacher-student support and student-student support but not opportunities for autonomy have the protective effect on the IGD through decreasing the chance of deviant peer affiliation.
Li, D., Zhou, X., Li, X., Zhou, Z. (2016). Perceived school climate and adolescent Internet addiction: The mediating role of deviant peer affiliation and the moderating role of effortful control. Computers in Human Behavior, 60, 54-61.
Ma, N., Zhang, W., Yu, C.-f., Zhu, J.-j., Jiang, Y.-p., & Wu, T. (2017). Perceived school climate and internet gaming disorder in junior school students: A moderated mediation model. Chinese Journal of Clinical Psychology, 25(1), 65–69.
Comment 8.The authors are encouraged to check the formatting of the references cited as there are inconsistencies throughout the list and with the requirements for the journal. It is important to update the references (last five years), some of them are much old.
Response: Thank you for your critical and careful review. We carefully checked and revise the reference format and replaced some old references.
We thank the editor and the reviewers for the constructive feedback, and we hope that you all find the current manuscript significantly improved. We look forward to your feedback on our revised paper.